# Experiments with Snails Add to Our Knowledge about the Role of aPKC Subfamily Kinases in Learning

**DOI:** 10.3390/ijms20092117

**Published:** 2019-04-29

**Authors:** Ekaterina Chesnokova, Alena Zuzina, Natalia Bal, Aliya Vinarskaya, Matvey Roshchin, Alexander Artyuhov, Erdem Dashinimaev, Nikolay Aseyev, Pavel Balaban, Peter Kolosov

**Affiliations:** 1Cellular Neurobiology of Learning Lab, Institute of Higher Nervous Activity and Neurophysiology, Russian Academy of Sciences, Moscow 117485, Russia; chesnokova@ihna.ru (E.C.); lucky-a89@mail.ru (A.Z.); bal_nv@mail.ru (N.B.); aliusha1976@mail.ru (A.V.); matvey.r87@gmail.com (M.R.); asenic@yandex.ru (N.A.); pmbalaban@gmail.com (P.B.); 2Department of Translational Medicine, Pirogov Russian National Research Medical University, Moscow 117997, Russia; alexanderartyuhov@gmail.com (A.A.); dashinimaev@gmail.com (E.D.); 3Laboratory of Cell Biology, Koltzov Institute of Developmental Biology, Russian Academy of Sciences, Moscow 119334, Russia

**Keywords:** atypical PKCs, PKMζ, mollusks, *Helix lucorum*, mRNA isoforms, mRNA expression, 5′-RACE, learning and memory

## Abstract

Protein kinase Mζ is considered important for memory formation and maintenance in different species, including invertebrates. PKMζ participates in multiple molecular pathways in neurons, regulating translation initiation rate, AMPA receptors turnover, synaptic scaffolding assembly, and other processes. Here, for the first time, we established the sequence of mRNA encoding PKMζ homolog in land snail *Helix lucorum*. We annotated important features of this mRNA: domains, putative capping sites, translation starts, and splicing sites. We discovered that this mRNA has at least two isoforms, and one of them lacks sequence encoding C1 domain. C1 deletion may be unique for snail because it has not been previously found in other species. We performed behavioral experiments with snails, measured expression levels of identified isoforms, and confirmed that their expression correlates with one type of learning.

## 1. Introduction

### 1.1. Atypical PKCs and Their Common Properties

Protein kinase C family of the serine/threonine protein kinases is divided into three subfamilies: classical, novel, and atypical PKCs. Subfamilies differ in their domain composition and ability to bind second messengers [1,2,3]. Kinases from atypical (aPKC) subfamily (the single aPKC isoform in invertebrates, PKCζ, PKMζ, and PKCι/λ in vertebrates) are not regulated by diacylglycerol (DAG) and Ca^2+^ like other PKC family kinases. The reason of this is that aPKCs lack the calcium-sensitive C2 domain and their C1 domain is not DAG-sensitive [1,4]. Important regulatory domains in most kinases from aPKCs subfamily are PB1 (protein interaction module) [5] and the pseudosubstrate motif. Pseudosubstrate segment occupies the active center of the enzyme and has to be relocated for kinase activation [6,7]. In this work, we focused on snail aPKC and its isoforms, and we compare them with homological kinases in vertebrates, PKCζ and PKMζ.

### 1.2. The Role of C1 Domain in Atypical PKCs Function

In atypical PKCs, the C1 domain is not able to bind DAG, but is still important for kinase function because it affects the subcellular localization of the molecule. In other PKC subfamilies, DAG interaction with C1 domain induces translocation of PKC to the plasma membrane. This triggers a conformational change that makes the pseudosubstrate to dissociate from the catalytic domain, resulting in kinase activation. The activation of atypical PKCs also involves plasma membrane association and release of the pseudosubstrate but does not require the DAG signal. It was shown that the same mutations in aPKCs C1 that prevent this domain from recognizing DAG also change the electrostatic potential of the domain and facilitate C1 interaction with negatively charged membrane. Moreover, the mutant form of C1 is more prone to nuclear translocation [4]. The nuclear localization signal is supposedly located in the N-part of C1 domain in mammalian aPKCs [8,9]. It was shown that isolated C1 domains of PKCζ and -ι tend to accumulate in the nucleus. They were also associated with the plasma membrane [4]. Full-size kinases have different properties compared to isolated domains because other domains affect the accessibility of the nuclear localization signal. Tagged full-size PKCζ was exclusively localized in the cytoplasm while tagged PKCλ was mostly cytosolic but was also present in nuclei. Nuclear import of PKCζ was much less efficient than that of PKCλ, as was shown in cells treated with an inhibitor of nuclear export. This result was unexpected considering that many known PKCζ targets are nuclear proteins [8]. It was later demonstrated that the ζ-specific hinge region prevents the nuclear import of PKCζ [9]. However, a simple inhibition of nuclear export is most likely not enough for PKCζ accumulation in the nucleus: some specific signal may be necessary for its active nuclear import. This has been already demonstrated for PKCɩ: phosphorylation of PKCɩ is needed for its import to the nucleus [10,11].

Graybill et al. [6] found that artificial deletion of C1 domain increases baseline activity of Drosophila’s aPKC. They suppose that C1 plays a critical role in synergizing with the pseudosubstrate to maintain the autoinhibited state. There are also a few reports about regulatory proteins binding C1 domain in aPKCs: LIP (lambda-interacting protein) and Par-4 (prostate androgen response-4) [12], but it seems like these proteins were not studied enough.

### 1.3. PKMζ Structure, Regulation, and Function

PKMζ is a very unusual molecule even when compared to other kinases of the same aPKC subfamily. Technically, PKMζ is a truncated form of PKCζ molecule. PKMζ only has the catalytic domain and lacks all upstream regulatory domains of PKCζ, so this small kinase is constitutively active [13]. Because PKMζ lacks all regulatory domains, the only way of controlling its activity is by changing its concentration in the cell. One of the proposed mechanisms of PKMζ regulation is its precisely controlled translation. It was demonstrated that the mRNA encoding PKMζ in mammals has extremely complicated structure of its 5′-UTR that inhibits its translation normally and allows it to accelerate in specific conditions associated with neuronal activation [14,15]. 

The function of PKMζ is strongly associated with memory formation and maintenance. Its role in these processes seems to be evolutionarily conserved, as experiments with inhibition or overexpression of PKMζ and its homologs showed similar results in mammals, mollusks, and insects [16]. In mammals, this unusual kinase is expressed exclusively in the central nervous system [13]. Within neurons, PKMζ immunoreactivity was detected not only in cell bodies and dendrites, but also in nuclei. Such distribution suggests the participation of this kinase in cell-wide mechanisms involving gene expression. Still, it is supposed that most mechanisms important for memory maintenance involve PKMζ molecules localized at postsynaptic densities and dendritic spines [17]. 

This kinase constitutive activity is believed to be necessary to maintain the facilitated synaptic connections that represent the memory engram in the brain [18,19,20]. In a computer model of biochemical network of PKMζ, it was demonstrated that PKMζ, being able to induce its own synthesis, acts as a bistable switch in neuron. That means the system responds to stimuli in an all-or-nothing manner, staying in “up” state indefinitely after reaching it once. This mathematical model conforms to the hypothesis of PKMζ having a pivotal role in long-term storage of memory. The model, despite being very simplified, was able to reproduce a variety of previous experimental results regarding synaptic plasticity and learning [21].

PKMζ participates in synaptic plasticity control on molecular level in many ways. Here we describe a few of the most important molecular pathways involving PKMζ.

Neuronal stimulation induces PKMζ translocation to the nucleus where it can affect transcription by direct phosphorylation of CREB-binding protein which subsequently increases the acetylation level of histones H2B and H3 [22]. 

Translation regulation by PKMζ may be realized via initiation factors eIF4B and eIF4E. PKMζ directly phosphorylates eIF4B, which causes the overall translation rate in the cell to decrease [23]. eIF4E is regulated indirectly, with PKMζ phosphorylating peptidyl-prolyl isomerase Pin1 that interacts with eIF4E, which eventually leads to the increase of the translation rate [24]. Thus, PKMζ is important for both translation suppression and translation activation mechanisms.

In the postsynapsic region, this kinase indirectly regulates AMPA receptors turnover. One of PKMζ substrates is PICK1, a protein that interacts with GluA2 subunit preventing its incorporporation into the postsynaptic membrane. PKMζ phosphorylates PICK1 that releases GluA2 [25]. The participation of PKMζ in postsynaptic site reorganization is also represented by PKMζ phosphorylation of ZDHHC8, palmitoyltransferase that promotes insertion of PSD-95 to the postsynaptic membrane [26]. 

Other putative substrates of PKMζ are proposed based on the similarity of PKMζ and PKCζ kinase domains but have yet to be confirmed experimentally. Since PKCζ is involved in the regulation of cell proliferation and survival [27], some of the currently known PKCζ substrates have very important functions in the cell. Among these substrates are, for example, Na,K-ATPase α-subunit [28], transcription factor Sp1 [8], RelA (a part of NF-κB transcription factor complex) [29], MEK kinase [30], Notch receptor [31], caspase 9 [32], and vesicle-associated membrane protein VAMP2 [33]. 

There is data confirming that PKMζ has catalytic activity very similar to PKCζ. PKCζ is known to phosphorylate kinase MARK2, causing its inactivation and translocation from the plasma membrane to the cytozol [34]. In the experiment performed by Tobias et al. [35], catalytic activity of exogenously expressed PKMζ and full-length PKCζ was compared. It was demonstrated that ability of both kinases to phosphorylate MARK2 and induce its translocation to the cytosol was similar. Moreover, in the same study it was proposed that PKMζ is able to phosphorylate multiple proteins inaccessible to PKCζ. These proteins were not specified but their number was assessed by Western blotting using a specific antibody for phosphorylated serine within common PKCs recognition site. Samples with PKMζ overexpression had more bands corresponding to different phosphorylated proteins [35]. 

### 1.4. Mechanisms of PKMζ Formation in Different Species

The structures of major subtypes of PKC family kinases are evolutionarily conserved throughout the animal kingdom, and aPKCs are supposed to diverge from the AGC branch of the human kinome earlier than other PKCs [2,3]. Nevertheless, mechanisms of exclusion of regulatory domains that are absent in PKMζ were shown to be different in different species [16]. 

In vertebrates, PKMζ is generated by alternative transcription. *Prkcz*, the gene that encodes PKCζ, has an alternative transcriptional start site corresponding to shortened mRNA that encodes PKMζ protein. PKCι-encoding gene does not have such alternative transcription start [36]. It was demonstrated that there is a specific promoter for PKMζ-encoding mRNA within the *Prkcz* gene. In rat, this promoter is normally active only in the brain [37]. Hernandez et al. [13] were the first to demonstrate that the mRNA transcribed from this promoter encodes protein in vitro. Using PKCζ-reg knockout mice (in which regulatory domain-encoding part of *Prkcz* gene was modified but catalytic domain-encoding part was intact) they also demonstrated that PKMζ may be generated in vivo in the absence of full-size PKCζ kinase. Homozygous PKCζ-reg knockout animals completely lacked PKCζ in cerebellum and kidney, but PKMζ in brain was preserved. It was shown that this internal promoter region is conserved in mouse, rat, and human. It contains several putative binding sites for activity-dependent transcription factors: CRE, NF-κB, and C/EBP.

In invertebrates, there is only one aPKC isoform that may be considered homological to both PKCζ and PKCι/λ. In *Aplysia californica*, a model invertebrate organism, PKMζ-like protein (independent kinase domain of aPKC) is generated posttranslationally as a result of aPKC proteolysis [36]. It was shown that Aplysia’s aPKC is cleaved in vivo during memory formation and that the requirements for such cleavage are the same as for neuronal plasticity [38]. 5′-RACE of Aplysia’s aPKC mRNA performed by Bougie et al. [36] did not reveal any alternative transcription start sites, but two alternative splicing sites were discovered. The two splice inserts encode the calpain cleavage site located in the hinge region between regulatory and catalytic domains of PKCζ. Without these inserts, the proteolysis was still possible but the cleavage site was different and the efficiency of the reaction was decreased.

Interestingly, the cleavage of PKCζ generating an independent catalytic domain was also demonstrated in mammalian cells, but only in the context of apoptosis. UV irradiation causes cleavage of PKCζ by a caspase-mediated process. The fragments generated by this cleavage are similar to catalytic domain but are enzymatically inactive, so in this case proteolysis is only one of the few mechanisms that inhibit PKCζ function in apoptosis [27]. It must be noted that there is no homology between Aplysia’s and mammalian cleavage sites. 

### 1.5. Land Snail as a Model Object for Neuroscience Studies

Mollusks provide an extremely useful model for deciphering the cellular mechanisms of behavior because their behavior is relatively complex, but their nervous system is quite simple and easily accessible for analysis. One of the best-known examples of neuroscience research performed in mollusks is Eric R. Kandel’s study. Using sea slug Aplysia, Kandel was the first to demonstrate how biochemical processes on the synaptic level are connected with changes in behavior [39,40].

Compared to Aplysia, terrestrial snails are mollusks that are relatively easy to keep in the lab. In their behavioral repertoire they have all major forms of animal behavior: feeding, avoidance, exploratory and sexual behavior. In our lab, experiments with snails have been performed for many years. The protocols for long-term sensitization, aversive conditioning, and even self-stimulation in snails were designed during this time [41,42]. Recently, we started to study neurons of snails on the molecular level and present here our first results. We established the sequence of aPKC-encoding mRNA in snail using two kinds of 5′-RACE (rapid amplification of cDNA 5′-end), discovered that this mRNA has at least two isoforms and measured their expression levels in naive and trained snails using RT-qPCR and droplet digital PCR.

## 2. Results

### 2.1. Classic 5′-RACE Revealed a Full-Size Transcript and a Few Shorter Fragments

Here we will compare our findings about snail’s aPKC mRNA with what is known about *Aplysia californica*’s aPKC mRNA. There is currently one confirmed sequence (NM_001204587.1) that encodes both full-size aPKC and truncated aPKC (PKMζ homolog) in Aplysia. There is also a predicted transcript variant of the same gene (XM_013088673). The predicted mRNA has a deletion. A fragment from G^784^ to G^857^ may be removed during splicing, and the PKCζ protein translated from such mRNA lacks 24 amino acids in the hinge region between C1 and catalytic domain. This fragment contains the cleavage site [36]. mRNA sequences of these two isoforms are present in Figure A1, translated protein sequences are present in Figure 1, and schematic depictions of protein structures are present in Figure 2a. We found two mRNA contigs in snail transcriptome assembly that bear homology to the Aplysia’s sequences described above. The shorter of two contigs has a 132-base deletion of the major part of C1 domain-encoding sequence. mRNA sequences of these contigs are present in Figure A1 in the Appendix B (and also in the Appendix A ), translated protein sequences are present in Figure 1, and schematic depictions of protein structures are present in Figure 2b.

For classic 5′-RACE, the agarose gel visualization of the third round of nested PCR revealed a mix of products (Figure A2a). We isolated a few prominent bands from the smear and cloned them. Two out of seven sequenced cloned products had exactly the same start position, so we supposed that it might be one of the possible transcription starts. Alignment of these cDNA sequences to the original snail contigs is presented in Figure A1, and schematic depictions of translated proteins are presented in Figure 3. In Figure A1 we show only two sequences: the longest product, presumably corresponding to the mRNA encoding the full-size aPKC protein (named “5’RACE full”), and one of the shorter products with the start position described above (named “5’RACE short”). Other products were aligned to the same sequence but had different starts, so they are not shown (schematic positions of fragment starts are marked on Figure 3). We suppose that some of the shorter products may correspond to the mRNA encoding the truncated aPKC protein, a homolog of mammalian PKMζ, and other shorter products may be artifacts of RNA fragmentation.

### 2.2. RLM-5’RACE Revealed Two Putative Capping Sites and Two Alternative Splicing Sites

Using RLM-5’RACE (RNA ligase-mediated rapid amplification of cDNA 5′-end), we were able to identify the capping sites of the examined mRNA. The agarose gel visualization of the second round of nested PCR in this experiment revealed three distinctive products, two bands with length approximately 500 and 650 bp from the first snail and one band with length approximately 250 bp from the other snail (Figure A2b,c). Subsequent sequencing of cloned recombinant plasmids confirmed that all the cloned products are indeed isoforms of the same mRNA sequence that is aligned to the snail contigs assembled earlier. One of the two isoforms with the same capping site had a 132-base insertion. The insertion bears homology to Aplysia’s annotated mRNA sequence, namely to a fragment that encodes a major part of C1 regulatory domain in Aplysia’s aPKC. This insertion was absent in one of the two snail contigs and in all classic 5′-RACE sequences. The same isoform had a 284-base deletion affecting the hinge region and the N-part of the kinase domain. We will call this isoform “aPKC X1” because Aplysia’s putative aPKC isoform X1 also has a deletion in the hinge. We will call the other isoform devoid of C1 domain “aPKCΔC1”. We supposed that the aPKC X1 mRNA isoform must be less abundant than the aPKCΔC1 one because most sequencing products did not have the C1 domain-encoding sequence. (ddPCR performed later confirmed this idea, as described below). The third isoform was the shortest and only had the sequence encoding the kinase domain, so we called it “aPKC KD”.

The original cDNA sequences of these isoforms are present in Figure A1, translated cDNA sequences are presented in Figure 1, and schematic depictions of translated proteins are presented in Figure 3. We suppose that two variants with the same transcription start, aPKCΔC1 and aPKC X1, are splice isoforms. Putative transcription starts of all three RLM-5’RACE isoforms are located downstream to the transcription start of putative full-size aPKC mRNA, and probably proteins encoded by RLM-5’RACE isoforms (especially by aPKC KD) may be considered variants of a truncated kinase similar to PKMζ in mammals. Still, we do not call them “PKM isoforms” below because primer pairs that we designed to distinguish cDNA corresponding to these isoforms may amplify cDNA corresponding to full-size aPKC mRNA variants as well.

We suppose that shorter transcripts that we discovered using the RLM-5’RACE approach represent capped mRNAs, so we tried to locate possible translation starts downstream of corresponding capping sites. However, in the raw sequencing data the remaining adapter sequence was not exactly like expected, with the last adapter base being absent in all plasmids. This one base was probably lost during the ligation of the adapter, but we were not able to find any data describing the mechanism of such a ligation error. The only other explanation we can provide is as follows. There is a homology between the 3′-end of the 5′-RACE inner primer from the kit (5′-CGCGGATCCGAACACTGCGTTTGCTGGCTTTGATG-3′) and the sequence that precedes both putative capping sites in the snail contig (TGATG). It is possible that the adapter ligation was unsuccessful, but due to this homology, 5′-RACE inner primer annealed to the middle of the template instead of the adapter. If this is what has happened, then we did not find the actual starts of capped sequences, but our discovery of two splice isoforms is still valid. It was confirmed later in experiments with specific primers to isoforms with or without 132-base insertion.

The shortest isoform, aPKC KD, has been only identified in one snail and was not found later in our quantification experiments with more snails, so the status of this fragment as an independent isoform remains the matter of doubt. 

### 2.3. The Training of Snails Was Successful in Both Experiments

#### 2.3.1. Contextual Fear Conditioning

The results of this experiment are presented in Figure 4a. The T0 testing session confirmed that prior to the conditioning, the experimental context, by itself, did not induce aversive reaction in snails. Before training, the mechanical stimuli caused snails’ posterior tentacles to shorten for 6.0 ± 1.6% of their initial length in the control context (on the glass) and for 6.9 ± 2.2% in the experimental context (on the floating ball), so initially both contexts were similar for snails. 

The training made snails to associate the experimental context with the electric shocks, and after the training the same mechanical stimuli applied to snails in the experimental context caused tentacle contraction for 57.0 ± 12.0% of their initial length (*p* < 0.001, Wilcoxon matched pairs test, compared to the T0 results on the ball). Training on the ball did not significantly affect tentacle contraction reaction to the touch observed in the control context: after the training, the measured parameter in the control context was 7.5 ± 1.7%.

#### 2.3.2. Taste Aversion Learning

The results of this experiment are presented in Figure 4b,c. The initial latency to touch the cotton bud measured during T0 was 19.8 ± 3.7 seconds, and it kept increasing each day of training. There was a highly significant dependence between the measured parameter and the number of training sessions (Friedman ANOVA Chi Sqr. (*n* = 4, df = 4) = 15.40000, *p* = 0.00394). After the training (during T1), the measured parameter reached 119.1 ± 1.8 seconds (*p* = 0.068, Wilcoxon matched pairs test, compared to the T0 result). All tested snails achieved the learning criterion.

### 2.4. ddPCR Demonstrated That the Ratio of Two Splice Isoforms in Subesophagial Ganglia is Roughly 2:1 and Does Not Change after Contextual Fear Conditioning 

Using ddPCR, we calculated the number of aPKCΔC1 molecules and aPKC X1 molecules in all examined samples. “aPKCΔC1 isoform copy number/aPKC X1 isoform copy number” ratio was 204 ± 58% in subesophageal ganglia from naive snails and 184 ± 26% in the ganglia from snails subjected to contextual fear conditioning. There was no significant difference between these two groups (Figure 5a–c, left and middle box plots).

For each examined sample we also calculated the total amount of aPKCΔC1 and aPKC X1 molecules taken together and divided this sum to the number of kinase domain-encoding fragments to calculate the percentage of the shortest isoform, aPKC KD, by a process of elimination. The result suggests that in the examined ganglia there are only two major isoforms. 95% confidence interval of “combined copy numbers of two isoforms/kinase domain copy number” ratio included “100%” value in both experimental groups and in the combined group as well (Figure 5a–c, right box plots). Thus, the shortest isoform is either extremely rare or even does not exist at all and was an artifact of RLM-5’RACE method.

### 2.5. qPCR Results Demonstrated That aPKC mRNA Expression Is Increased in Snail Ganglia after Taste Aversion Learning, but Not After Contextual Fear Conditioning

Contextual fear conditioning did not have any significant effects on relative expression levels of total aPKC or either of its splice isoforms. However, in snails that were subjected to taste aversion learning we detected upregulation of total aPKC-encoding mRNA and of both its isoforms (Figure 6a–c). In subesophageal ganglia, the relative increase of total aPKC-encoding mRNA expression level after training was 160 ± 35% compared to the control group, the increase of aPKCΔC1 isoform expression was 177 ± 27%, and for the aPKC X1 isoform it constituted 163 ± 38% (in all cases, *p* < 0.05, Mann–Whitney test). In pedal ganglia, the increase was slightly less prominent: 138 ± 14% for total aPKC-encoding mRNA (*p* < 0.05, Mann–Whitney test), 127 ± 13% for aPKCΔC1 isoform (*p* < 0.05, Mann–Whitney test), and 160 ± 50% for aPKC X1 isoform (*p* = 0.073, Mann–Whitney test).

Since some snails were unable to learn and were excluded from the taste aversion experiment, there is a possibility that more “clever” snails, which comprised the trained group, had inherently increased aPKC expression. In this case, increased aPKC expression is important for learning but may not be induced by it. We tried to evaluate the impact of training using statistics. Relative gene expression levels usually have lognormal distribution (the logarithms of values are distributed normally [43]). We calculated logarithms of expression levels of all targets for all samples and confirmed that these variables are distributed normally in the control (naive) group using Shapiro–Wilk test. We then calculated AM + 3 SD (average mean + 3 × standard deviation; upper three sigma limit, or 99.87th percentile) for every distribution in the control group. For every snail in the trained group, we compared logarithms of expression levels of every target with AM + 3 SD in corresponding control group (data not shown). For total aPKC-encoding mRNA and aPKCΔC1 isoform, all values for trained snails were below AM + 3 SD in corresponding control group, so the training did not cause expression levels increase so much that they are outside of the distribution characteristic for naive snails. However, in two out of four pedal ganglia of trained snails logarithm of aPKC X1 isoform relative expression was higher than AM + 3 SD for this parameter in control. Therefore, we can suppose that such levels of aPKC X1 isoform in pedal ganglia are not natural for naive snails and represent the increase of expression caused specifically by taste aversion training.

To compare expression levels of each isoform between experimental groups, we used qPCR and not ddPCR because qPCR is cheaper and easier to perform so we could test more samples. Still, for 10 samples that were used in both qPCR and ddPCR, expression fold changes measured by ddPCR (data not shown) were very similar to the values calculated using qPCR.

## 3. Discussion

Despite the land snail being a model object for electrophysiological and behavioral experiments, the molecular biology of this species is not yet studied very well. Very few molecular neuroscience experiments in snail have been reported before. One of the most suitable references that we had was the annotated mRNA sequences from *Aplysia californica*, so we base our conclusions on what is known about Aplysia’s aPKC and corresponding mRNA. In Aplysia, the same mRNA encodes both full-size aPKC and truncated aPKC, and this mRNA has two splice isoforms [36]. We find it plausible that *Helix lucorum* should also have two aPKC kinases—full-size and truncated—because they are both important and evolutionarily conserved. We also suppose that in snail, like in other species, these two kinases should be encoded by the same mRNA. We found two different mRNA contigs in snail transcriptome assembly that bear homology to the Aplysia’s reference mRNA (Figure A1; Appendix A ). The shorter of two contigs has a deletion in the C1 region.

In the classic 5′-RACE experiment, we identified only sequences corresponding to the shorter contig (so we called this isoform aPKCΔC1). This isoform was also present in RLM-5’RACE experiment results and its existence was later confirmed using PCR methods. As far as we know, deletion of C1 domain in aPKC was never described earlier for any other organism. It is known that C1 domain in aPKCs is DAG-insensitive, but its electrostatic properties are important for both plasma membrane association and nuclear translocation of the molecule [4]. Knowing this, we may speculate that snail aPKCΔC1 protein most likely has cytosolic localization. Still, this kinase isoform contains six out of seven amino acids (A^132^–K^138^) that may serve as a nuclear localization signal in snail (based on comparison with the mammalian sequence [9]) and are located just upstream of the missing C1 sequence (Figure 1). Considering that an artificial deletion of this domain in Drosophila’s aPKC performed by Graybill et al. [6] resulted in increased catalytic activity, we may also suppose that aPKCΔC1 kinase isoform in snail is more active than aPKC X1 described below. Snail aPKCΔC1 isoform contains the sequence that corresponds to the splice inserts in Aplysia’s aPKC mRNA. These inserts encode the main cleavage site in aPKC protein [36]. This means that the mechanism of PKMζ-like protein formation by cleavage demonstrated in Aplysia may work in land snail too. Still, it is questionable if in snail this cleavage site is functional. The homology between snail and Aplysia’s protein sequences in this region is only 63% (compared to 79% overall homology between full Aplysia’s aPKC protein and sequence translated from our snail contig without C1 domain), and there are 29 amino acids in this region of snail protein instead of 25 in Aplysia’s protein. 

Another isoform that has the insertion corresponding to the C1 domain in the longer contig was found only in the RLM-5’RACE experiment, but its existence was later confirmed with PCR. The same isoform also had a deletion in the region encoding the hinge between C1 and catalytic domain. Because of this, we decided to call this snail isoform aPKC X1, like Aplysia’s predicted aPKC mRNA isoform X1 with a deletion in the hinge. In snail’s aPKC X1 this deletion is very large, so the protein it may encode will lack a major part of the hinge and the first 15 amino acids out of 328 constituting catalytic domain. Without conducting specific biochemical experiments, it is impossible to know for sure if such a deletion will affect the catalytic function of the kinase. However, based on the literature data about PKC tertiary structure, these 15 amino acids are supposedly not crucial for the catalytic activity or for the folding of protein core [44]. aPKC X1 isoform has a C1 domain characteristic for aPKCs (DAG-insensitive). Based on what is known about the function of this domain, it is most likely that aPKC X1 baseline activity is decreased [6]. It was shown that in human, the difference between atypical (PKCζ) and novel (PKCδ) C1 domains is 4 point mutations changing amino acids to positively charged arginine residues. This changes the electrostatic profile of this domain and favors its interactions with negatively charged membranes [4]. The 132-base insertion in C1 domain of snail aPKC X1 isoform adds 44 amino acids to the translated protein, with four of them being arginines and six being lysines that are positively charged (Figure 1). All this allows us to propose that this isoform has specific electrostatic properties. We hypothesize that the atypical C1 domain in snail’s aPKC X1 may be important for its plasma membrane association. It may also be necessary for transportation of the kinase to the nucleus or for specific substrates binding, as was proposed for other species [4]. Probably there are also some regulatory proteins able to interact with this domain, as was reported earlier for other species [12]. aPKC X1 does not have regions corresponding to either Aplysia’s calpain cleavage site [36] or mammalian caspase cleavage sites [27], so most likely aPKC X1 cannot produce a PKMζ-like protein by proteolysis.

Strangely, the shortest isoform, aPKC KD, was revealed in the RLM-5’RACE experiment but was not later detected using ddPCR approach: all the molecules that had the sequence encoding kinase domain also had upstream sequence encoding either the 132-base insertion or fragments surrounding the insertion (Figure 5). So we suppose that the shortest isoform is either extremely rare or was identified erroneously. We used two different snails for RLM-5’RACE experiment, and only one of them had the shortest isoform; moreover, the same snail did not have two longer isoforms (Figure A2). Probably this specific snail had a rare mutation. We used relatively large sample group for ddPCR (10 snails), so we believe that ddPCR results are more representative. In addition, shorter products have advantage in amplification rate, and it may explain why we did not amplify longer isoforms in the sample from the mutated snail.

We propose two alternative translation starts for shortened isoforms of snail aPKCs based on translation frame, domains disposition, and locations of putative capping sites (Figure 1, Figure 2 and Figure 3, Figure A1). Both of these starts are absent in Aplysia’s sequences. It must be noted that a separate PKMζ translation start confirmed in vertebrates is located right after C1 domain and is absent in both snail and Aplysia [36]. As was mentioned above, it is possible that we did not locate the capping sites correctly. It may explain why aPKC KD isoform was not detected in later experiments. In this case, it is possible that the same full-size RNA encodes all isoforms of aPKC in snail, and the truncation happens after translation, as it does in Aplysia. The exact mechanism of PKMζ-like protein formation in snail is yet to be investigated. However, our quantitative PCR experiments definitely confirmed the existence of two aPKC mRNA isoforms in snail that both have the catalytic domain but differ by 132-base fragment corresponding to the C1 domain. We are almost positive that they are splice isoforms, but they also may be products of two homological genes. More studies are necessary to exclude this possibility. 

Our behavioral experiments demonstrated that expression of both confirmed isoforms increases after taste aversion learning, but not after contextual fear conditioning in snails. This may indicate that these two types of learning are realized using different molecular pathways. There is not much known about signal pathways within snail neurons, but their electrophysiological properties and responses to some neurotransmitters have been thoroughly studied earlier in our lab. In experiments assessing the role of serotonin in snail CNS it was confirmed that associative (taste aversion) and environmental (contextual fear conditioning) types of learning indeed differ on the cellular level. It was demonstrated that serotonin is necessary for taste aversion learning only during the consolidation phase, but not for memory retrieval [45,46]. On the contrary, in experiments with contextual fear conditioning it was shown that serotonin is necessary for recall and/or retention of the obtained memory [46]. It was proposed that the information about certain specific (cued) stimuli that are connected with reinforcement, and the information about specific context are stored independently in the snail CNS, and this makes snail behavior more adaptive [46].

In our study, expression of both isoforms increased after training. It means that both are important for memory formation, but we did not find any indications of different functions of these isoforms yet. The ratio between aPKC X1 and aPKCΔC1 isoforms also seems to be stable and does not change when neurons are activated. In addition to the experiments described above, we also performed a pilot experiment with a few subesophageal ganglia that were isolated and electrochemically facilitated in vitro, and measured the expression level of aPKC isoforms in these ganglia (data not shown). The results of ddPCR revealed that these ganglia had the same isoform ratio as ganglia extracted from naive and trained snails.

Increased aPKC expression correlates with successful taste aversion training, so aPKC must be important for this kind of learning in snail. However, it is impossible to establish what is the cause and what is the effect in this correlation because we cannot analyze the ganglia of the same snail both before and after training. In model organisms like mice, genetic engineering can be applied to change expression level of genes in question directly and determine the effects of it, but very little is known about *Helix lucorum* genetics (there is not even an annotated snail genome yet), so the generation of genetically engineered snails is currently impossible. Still, we can speculate that the training can induce at least aPKC X1 expression in pedal ganglia, based on the fact that for this isoform some values in trained group lied outside the distribution characteristic for naive snails. In addition, in a few snail ganglia that were isolated and facilitated in vitro, we observed moderate increase of aPKC mRNA expression after the stimulation compared to the control isolated ganglia (data not shown). This result supports the hypothesis that aPKC mRNA expression is stimulated by neuronal activation rather than being perpetually increased in a small population of “clever” snails. Still, more research is necessary to study the nature of the correlation between training performance and aPKC mRNA expression in snail CNS.

We hope that our pioneer study will be followed by more molecular biology experiments with *Helix lucorum*. This animal was used as a model for neuroscience research for many years, and better understanding of behavioral and electrophysiological results accumulated during this time may be achieved if we know more about transcription and translation regulation in snail neurons.

## 4. Materials and Methods

### 4.1. Animals

Animal studies were approved by the Ethical Committee of the Institute of Higher Nervous Activity and Neurophysiology (Protocol #012, 12.02.2014). For our experiments, we used wild-caught adult land snails of *Helix lucorum* (Linnaeus, 1758) species (Crimea population). Snails were put in the fridge for hibernation for a few months and later caged in terrarium in glass boxes with high humidity conditions, enough space, food (cabbage), and water ad libitum. All the animals had similar size and weight. Before the experiments, snails were allowed to recover from hibernation for at least 2 weeks and gain some weight. In both behavioral experiments, snails were deprived of food 3 days before the experiment and during the whole experiment.

For 5′-RACE experiments, 4 naive snails were used (one was later excluded). For contextual fear conditioning, 13 snails were trained. For taste aversion learning, 4 snails were trained. Initially training has been started with more snails, but some of them were unable to learn and were excluded from experiments. In both behavioral experiments, we compared reactions of the same animal before and after training. Ganglia extracted from 14 naive snails (awakened from hibernation at the same time as the experimental snails) were later used as a control for PCR experiments. The term “subesophageal ganglia” here means the complex consisting of two parietal ganglia, two pleural ganglia and one visceral ganglion.

### 4.2. 5′-End of Snail aPKC mRNA Cloning and Sequencing

#### 4.2.1. Primer Design

*Helix lucorum* brain transcriptome assembly was made earlier by our collaborators—P. Khaitovich’s group. Sequencing of snail mRNA was performed using the HiSeq Illumina platform, and proteomes of five animal species from different phyla were used as references for de novo transcriptome assembly. The detailed description of this work will be presented in the paper of P. Khaitovich with coauthors that was recently accepted for publication [47].

Two specific contigs from snail transcriptome assembly that had significant homology to aPKC mRNA of other species (Figure A1; Appendix A) were used as references for primer design. Design of gene-specific primers was made using Clone Manager 7 software (http://www.scied.com/). Primer sequences are present in Table 1.

#### 4.2.2. RNA Extraction

Total RNA was extracted from the isolated CNS of snails with ExtractRNA reagent (Evrogen, Moscow, Russia) according to the manufacturer’s protocol. RNA quality was checked by spectrophotometry on NanoDrop 2000 (Thermo Fisher Scientific, Waltham, MA, USA). Purified RNA was dissolved in MQ water and stored at −70 ℃.

#### 4.2.3. Classic 5′-RACE cDNA Preparation and Amplification

5′-RACE (rapid amplification of cDNA 5′-end) is a method that allows amplification of an unknown sequence on the 5′-end of a specific mRNA, using a primer designed to anneal to a known mRNA part and a universal adaptor primer. RNA extracted from the CNS of a single naive snail was used in this experiment. We used the SMART approach [48,49] to prepare first-strand cDNA and anchor it with the adaptor sequence in a single step. cDNA synthesis was followed by Step-Out RACE procedure (a specific variant of nested PCR) [50] to amplify the 5′-end of the sequence. Reverse transcription paired with flanking of cDNA with adaptor sequences was performed with Mint cDNA synthesis kit (Evrogen, Moscow, Russia) according to the manufacturer’s protocol. cDNA was pre-amplified with a single adaptor-specific primer according to Evrogen’s protocol before being used as a template for nested PCR. Three rounds of nested PCR were performed with Mint RACE primer set and Encyclo polymerase mix (Evrogen, Moscow, Russia) using PCR programs recommended by Evrogen. Primer annealing temperature was 62 °C in every reaction. PCR products were visualized using 1% agarose gel electrophoresis.

#### 4.2.4. RLM-5’RACE cDNA Preparation and Amplification

The RLM-RACE method (RNA ligase-mediated rapid amplification of cDNA ends) allows amplifying cDNA ends selectively from full-length, capped mRNAs [51,52]. Two RNA samples extracted from the CNS of two naive snails were used in this experiment. To perform RLM-5’RACE, we used FirstChoice RLM-RACE Kit (Ambion, Inc., Foster City, CA, USA). Capped mRNA selection (removal of free 5′-phosphates from all noncapped molecules with calf intestine alkaline phosphatase, followed by treatment with tobacco acid pyrophosphatase that removes the cap), adaptor ligation to intact 5′-phosphates of selected molecules, reverse transcription and amplification were done according to the manufacturer’s protocol. First strand of cDNA was used as a template for nested PCR. Two rounds of nested PCR were performed with Q5 high-fidelity DNA polymerase (New England Biolabs, Ipswich, MA, USA) using PCR program recommended by Ambion and primer annealing temperature 62 °C in both reactions. PCR products were visualized using 1% agarose gel electrophoresis.

#### 4.2.5. Molecular Cloning and Sequencing

This procedure was the same for classic 5′-RACE and RLM-5’RACE products. PCR products of the last round of RACE were purified from gel with Wizard SV Gel and PCR Clean-Up System (Promega, Madison, WI, USA). A short amplification (7 cycles) of purified products with Taq-polymerase and the same primers and PCR cycling profile as in the last round of RACE was performed to add cohesive ends. After this, PCR products with cohesive ends were purified from gel one more time and ligated overnight to pAL2-T plasmid vector (Evrogen, Moscow, Russia) using T4 ligase (NEB) (PEG-4000 was added to the ligation mixture to the final concentration of 10%; after 30 min on RT ligation mixture was placed at 4 °C). The top 10 *E.coli* competent cells (NEB) were transformed with ligation products via heat shock and seeded on LB agar with added IPTG (1 mM) and X-Gal (10 μg/mL) for blue-white screening. After overnight incubation at 37 °C, a few white colonies were selected and additionally checked with colony PCR. Identified positive colonies were seeded in liquid LB and incubated overnight at 37 °C in a shaking incubator. Plasmid DNA was extracted using rapid alkaline extraction procedure [53]. Sanger sequencing of purified plasmids in both directions using standard sequencing primers T7 (in both cases), SP6 (for classic 5′-RACE) and M13_rev (for RLM-5’RACE) was performed by Evrogen company (Moscow, Russia).

#### 4.2.6. Alignment

Raw sequencing data were edited in SnapGene software (https://www.snapgene.com/). Vector and primers sequences were removed. Sequencing peaks were assessed visually and some bases were corrected manually. After this, sequences were aligned to snail’s putative aPKC contigs using NCBI-BLAST online service and SnapGene application. Annotated sequence of mRNA encoding aPKC kinase in *Aplysia californica* (NCBI Reference Sequence: NM_001204587.1) and the corresponding protein sequence (NCBI Reference Sequence: NP_001191516.1) were used to predict the positions of open reading frames and mRNA sequences encoding different protein domains in snail.

### 4.3. Behavioral Methods

#### 4.3.1. Contextual Fear Conditioning

Aversive reaction estimation: We scored the amplitude of the snail’s response to the moderate tactile stimulation to measure aversive behavior. Punctate mechanical stimuli were applied with calibrated von Frey hairs (pressure 25g/mm^2^) to the area located a few millimeters behind the left posterior tentacle. The pressure that we used was not painful and normally caused the tentacles to shorten for 5–10% of their initial length. Posterior tentacle contractions caused by the touch were filmed and the relative change in tentacle length was later measured in the video frames (the pretest value taken as 100%). The tentacle contraction in response to the mechanical stimulation was measured for each snail in two different contexts: on the flat glass surface (similar to the surface of snail’s home cage) or on the foil-covered ball described below. The reaction was measured 3 times in a row with 15-min intervals between trials.

Conditioning procedure: In the experimental setup, the snail was tethered by its shell in a manner allowing it to crawl on a ball that rotated freely in water containing 0.01% NaCl (details in [42,54]). The ball was covered with aluminum foil to make it conductive. To induce the contextual fear memory, electric shocks (1 second) were delivered via the metal electrode that was applied manually to the dorsal surface of the snail’s foot. The second (carbon) electrode was submerged in the solution in which the ball floated. The current was adjusted individually to make the snail retract its head and the anterior part of its leg into its shell, and varied from 1 to 4 mA for different snails. Every snail had at least 2 shocks each day of learning, with the interval between shocks being at least 1 h (during these intervals snails were removed from the experimental set and put in a container with increased humidity to keep them active). After a few days of learning, less current was necessary to provoke the reaction, so it was decreased. 

Experiment schedule: For the first two days of the experiment, snails were accommodated to the experimental set-up: they were put on the floating ball for 30 min each day. No electric shocks were administered in these two days. After 30 min on the ball, snails’ aversive reaction to the mechanical stimuli was measured in both contexts (initial testing, T0). After the accommodation, snails were trained for 10 consecutive days: every day they were put on the floating ball and received electric shocks strong enough to cause head and foot retraction response. After 10 days of training and one day of rest, another testing was performed (testing after learning, T1). Animals were sacrificed and ganglia were extracted at least 24 h after T1.

Statistical analysis of experiment results (Wilcoxon matched pairs test) was performed using STATISTICA 8 application (http://www.statsoft.com/Products/STATISTICA-Features).

#### 4.3.2. Taste Aversion Learning

Aversive reaction estimation: We used carrot smell (presented via a cotton bud soaked in carrot juice) as a neutral stimulus. The latency of snail touching the bud with its tentacles (consummatory response) was measured during every presentation of the smell (in training and testing sessions). The maximal time of odor presentation was limited to 120 s for each trial, and if the snail did not touch the bud after this time, it was interpreted as a refusal. For training and both testing sessions, snails were put in the experimental context (on the floating ball in the setup described above).

Aversion learning procedure: The carrot juice-soaked cotton bud was placed in front of lower tentacles, at the distance of 5 mm from them. Once the snail touched the bud, an electric shock was administered in a manner described above. Each shock lasted 0.5 s, and the current range was from 1 to 10 mA for different snails (enough to make a snail to withdraw its tentacles, to retract the anterior part of its body into its shell and to secret some mucus). If the snail refused to touch the bud for 120 s, no shock was applied. Combined presentations of neutral (carrot smell) and unconditioned (electric shock) stimuli were repeated 5 times each day of learning, with 15–20 min between trials.

Experiment schedule: Before the training, the initial testing session (T0) was performed: the latency of the consummatory response was measured for every snail 3 times, with 5-min intervals between trials. On the next day, the training started and continued for 5 days, with 5 trials per day. The training was considered to be successful (learning criterion achieved) if 4 consecutive refusals were observed. Testing after learning (T1) was performed at 24 h after the last learning session. Snails were presented with carrot smell 3 times with 5-min intervals between trials. If the snail touched the bud during T1, no shock was administered. Animals were sacrificed and ganglia were extracted at least 24 h after T1.

Statistical analysis of experiment results (Wilcoxon matched pairs test and Friedman ANOVA) was performed using STATISTICA 8 application.

### 4.4. Quantitative Assessment of Expression of Putative Snail aPKC mRNA Isoforms

#### 4.4.1. RNA Extraction and Reverse Transcription

Total RNA was extracted from snail CNS samples using ExtractRNA reagent (Evrogen, Moscow, Russia). Residual genomic DNA was removed from RNA samples with DNAse I (Life Technologies, Carlsbad, CA, USA), then DNAse I was inactivated by adding EDTA. Reverse transcription was then performed with MMLV reverse transcriptase (Evrogen, Moscow, Russia) and random decamer primers. All these reactions were performed according to the protocols provided by manufacturers. cDNA was not purified, but we diluted it before use for qPCR to prevent PCR inhibition by remaining EDTA and DTT. Part of each sample after DNAse reaction was used as “RT− control“ (no-reverse transcription control: instead of reverse transcription reagents, the same volume of MQ water was added to these controls to match gDNA dilutions in RT− and RT+ samples). Along with experimental samples (subesophageal and pedal ganglia), we extracted and reverse transcribed RNA from the reference sample (CNS fragments obtained from a few naive snails and combined together) to use it for positive control and calibration.

#### 4.4.2. Primer Design

To measure expression levels of aPKC mRNA isoforms, we designed 6 primer pairs for 5 targets. Primer sequences are present in Table 2. We made primers specific for each splice isoform identified by 5′-RACE (see Results): one primer specific for X1 isoform annealed on the 132-base insert, while one primer specific for ΔC1 isoform annealed on the nucleotides located on both sides of the insertion site. We also made a primer pair specific to a sequence encoding kinase domain that is common for all isoforms. For normalization purposes, we also designed primers for mRNAs of two housekeeping genes: TBP and GAPDH. To find putative TBP and GAPDH mRNA sequences in our *Helix lucorum* transcriptome assembly, we used the same strategy as with aPKC, comparing our contigs to the known sequences of mRNAs encoding these proteins in other mollusks (namely, *Crassostrea gigas* and *Perna canaliculus*). Primer design was performed using Clone Manager 7. Specificity of the primers was confirmed using NCBI Primer-BLAST service. Before using the designed primers for the experimental samples, we tested them empirically with positive (reference sample) and negative (no target) controls to choose the optimal annealing temperature and to confirm that primers do not amplify any unspecific byproducts. Based on the results of these pilot experiments, different primer pairs targeting aPKCΔC1 were selected for ddPCR and qPCR (we used different conditions for these two reactions). We also confirmed experimentally with agarose gel electrophoresis that the primers we used produce amplicons of expected lengths.

#### 4.4.3. Droplet Digital PCR

We performed ddPCR in addition to classic qPCR because ddPCR makes it possible to calculate the number of cDNA molecules of each kind per microliter directly, without complicated calibration and nonlinear errors [55,56], so we were able to calculate the isoform ratio. We used ready master mix QX200 ddPCR EvaGreen supermix (Bio-Rad, Hercules, CA, USA). The volume of initial PCR mix before droplet generation was 22 μL, the concentration of each primer in this mix was 0.09 μM, and the final dilution of cDNA matrix was 1:7.3. Along with experimental samples, we also amplified no-template (MQ) controls and no-reverse transcription (RT−) controls. Droplets were made on AutoDG Automated Droplet Generator (Bio-Rad) with default settings using QX200 droplet generator oil for EvaGreen dye (Bio-Rad). The amplification was performed on C1000 Touch Thermal Cycler (Bio-Rad) using the following PCR program.
Initial denaturation: 95 °C, 5 min, ramp rate 2 °C/s40 cycles of amplification: Denaturation: 95 °C, 30 s, ramp rate 2 °C/sAnnealing: 55.9 °C, 30 s, ramp rate 2 °C/sElongation: 72 °C, 30 sSignal stabilization: 4 °C, 5 min90 °C, 5 minInfinite hold, 12 °C

The microfluidic analysis of droplets fluorescence was performed using QX200 droplet reader (Bio-Rad) and calculation of droplets was made with QuantaSoft Software (http://www.bio-rad.com/en-us/sku/1864011-quantasoft-software-regulatory-edition?ID=1864011) with default settings.

#### 4.4.4. Droplet Digital PCR Analysis and Calculations

The threshold was set manually in QuantaSoft Software on 1D Amplitude graphs so all the wells with the same target had the same threshold and it was equally far from “positive” and “negative” peaks in most of these wells. The software then calculated the number of target molecules in each well. These numbers were used for statistical analysis (Mann–Whitney test, performed in STATISTICA 8 application).

#### 4.4.5. Quantitative PCR

For qPCR, we used ready master mix qPCRmix-HS SYBR+LowROX (Evrogen, Moscow, Russia). Each sample was amplified in triplicate. We also amplified no-template (MQ) controls, no-reverse transcription (RT−) controls and positive controls of 3 different cDNA concentrations (dilutions of the reference sample). The volume of each reaction was 12 μL, the concentration of each primer was 0.4 μM, and the final dilution of cDNA matrix was 1:96. The amplification was performed on CFX386 PCR machine (Bio-Rad) using the following PCR program.
Initial denaturation: 95 °C, 5 min40 cycles of amplification: Denaturation: 95 °C, 30 sAnnealing: 63 °C, 30 sElongation: 72 °C, 30 s, with detection of fluorescenceMelt curve: 65–95 °C, ramp rate 0.5 °C/5 s, with continuous detection of fluorescence

#### 4.4.6. Quantitative PCR Analysis and Calculations

Melt curve analysis was used each time to confirm reaction specificity. Reaction efficiencies were calculated using calibration curves. The calibration curve was plotted for each primer pair using amplification of 3 different dilutions of reference sample. We repeated calibration in each PCR plate. Threshold value was set manually every time as the lowest fluorescence intensity at which all amplification curves in log view look parallel. Background cycles were set manually, with the last background cycle being at least 2 to 3 cycles below C_t_ for the most concentrated sample. Some wells were omitted from the analysis manually based on the shape of amplification curves. The average mean of C_t_ for triplicates and reaction efficiency values were used to calculate relative cDNA concentration in each sample using Pfaffl method [57]. The concentration of cDNA in the reference sample was considered equal to 100% for each target, and all concentrations in other samples were calculated relative to this reference value. To normalize for total cDNA quantity, we measured relative concentrations of cDNA of two housekeeping genes and calculated their geometrical mean. Then we divided the relative quantity of kinase isoforms to this geometrical mean. These normalized numbers were used for statistical analysis (Mann–Whitney test, performed in STATISTICA 8 application). To visualize data for plotting graphs, all relative quantity values were additionally normalized to the average relative value in the control group. 

## Figures and Tables

**Figure 1 ijms-20-02117-f001:**
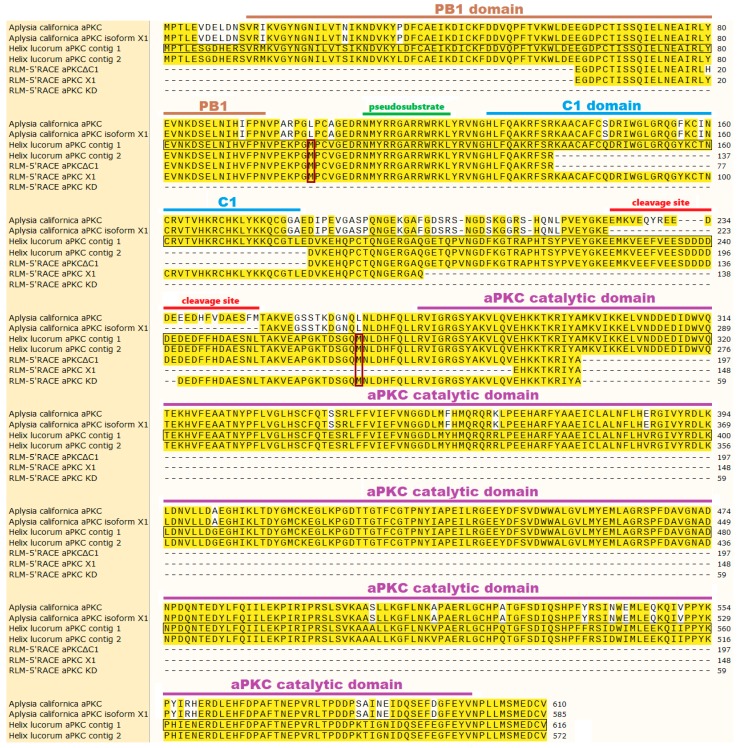
Multiple alignment of translated sequences: two isoforms of Aplysia’s aPKC (*Aplysia californica* aPKC and *Aplysia californica* aPKC isoform X1), two snail mRNA contigs (assembled from snail full-transcriptome sequencing data) homological to Aplysia’s aPKC mRNA (*Helix lucorum* aPKC contigs 1 and 2), and three RLM-5’RACE sequences (RLM-5’RACE aPKCΔC1, RLM-5’RACE aPKC X1, and RLM-5’RACE aPKC KD). Yellow coloring represents similarity to the common reference sequence, *Helix lucorum* aPKC contig 1 (framed). Domains and features are labeled based on similarity with annotated Aplysia’s sequences [36]. Red frames mark two possible translation starts present only in snail sequences.

**Figure 2 ijms-20-02117-f002:**
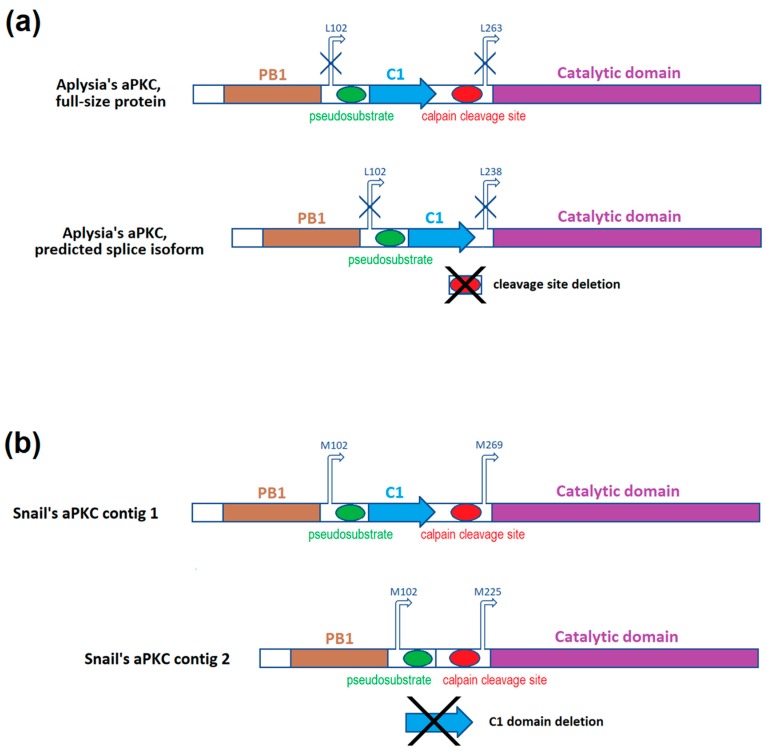
(**a**) Domains and features of two isoforms of Aplysia’s aPKC kinase (based on the paper by Bougie et al., 2009 [36]); (**b**) Domains and features of two *Helix lucorum* aPKC contigs (assembled from snail full-transcriptome sequencing data) based on similarity with Aplysia’s sequences. Arrows above the molecule represent possible translation starts (present only in snail sequences; corresponding sites in Aplysia’s sequences are marked with crossed arrows).

**Figure 3 ijms-20-02117-f003:**
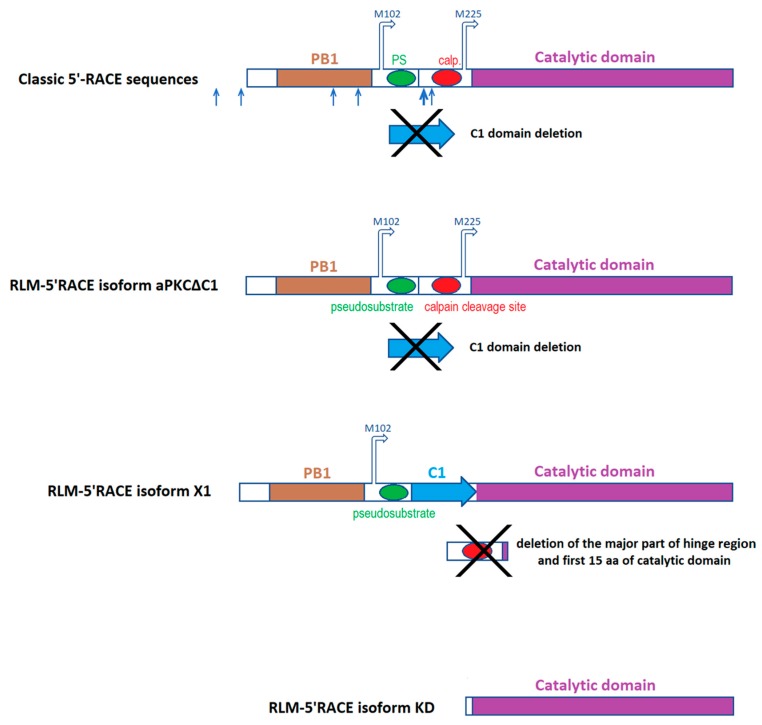
Schematic representations of putative snail aPKC isoforms based on 5′-RACE results. Arrows above the molecule represent possible translation starts. Numbers of amino acids in 5′-RACE isoforms are given based on corresponding snail contig. Arrows below the molecule in classic 5′-RACE sequences represent starts of fragments with different lengths (the largest arrow represents the common start of 2 fragments). PS–pseudosubstrate, calp.–calpain cleavage site.

**Figure 4 ijms-20-02117-f004:**
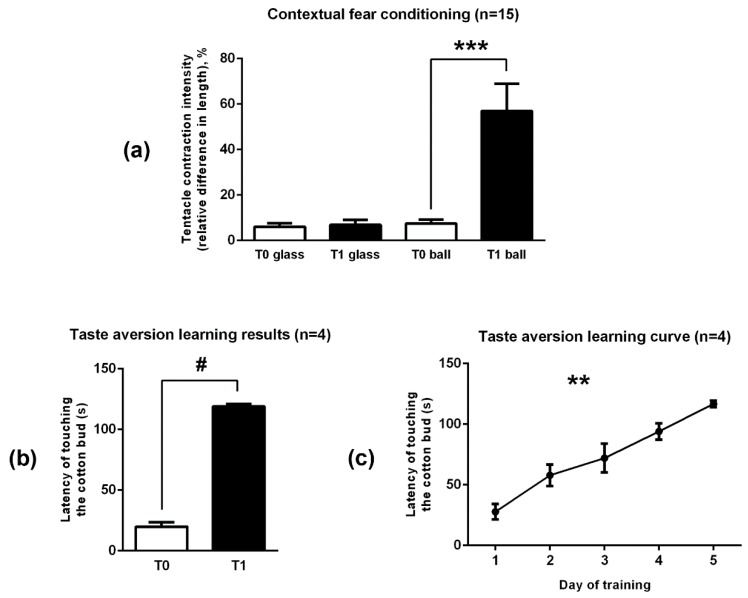
Snail training results: (**a**) Tentacle contraction intensity (relative length decrease) caused by the same mechanical stimuli applied to snails in control (glass) and experimental (ball) context before (T0) and after (T1) contextual fear conditioning. (**b**) Latency of the consummatory reaction to the presented carrot smell before (T0) and after (T1) taste aversion learning. (**c**) Latency of the consummatory reaction for every day of training during taste aversion learning. Data are present as AM ± SD. ** *p* < 0.005, Friedman ANOVA, *** *p* < 0.001, Wilcoxon matched pairs test, # *p* = 0.068, Wilcoxon matched pairs test.

**Figure 5 ijms-20-02117-f005:**
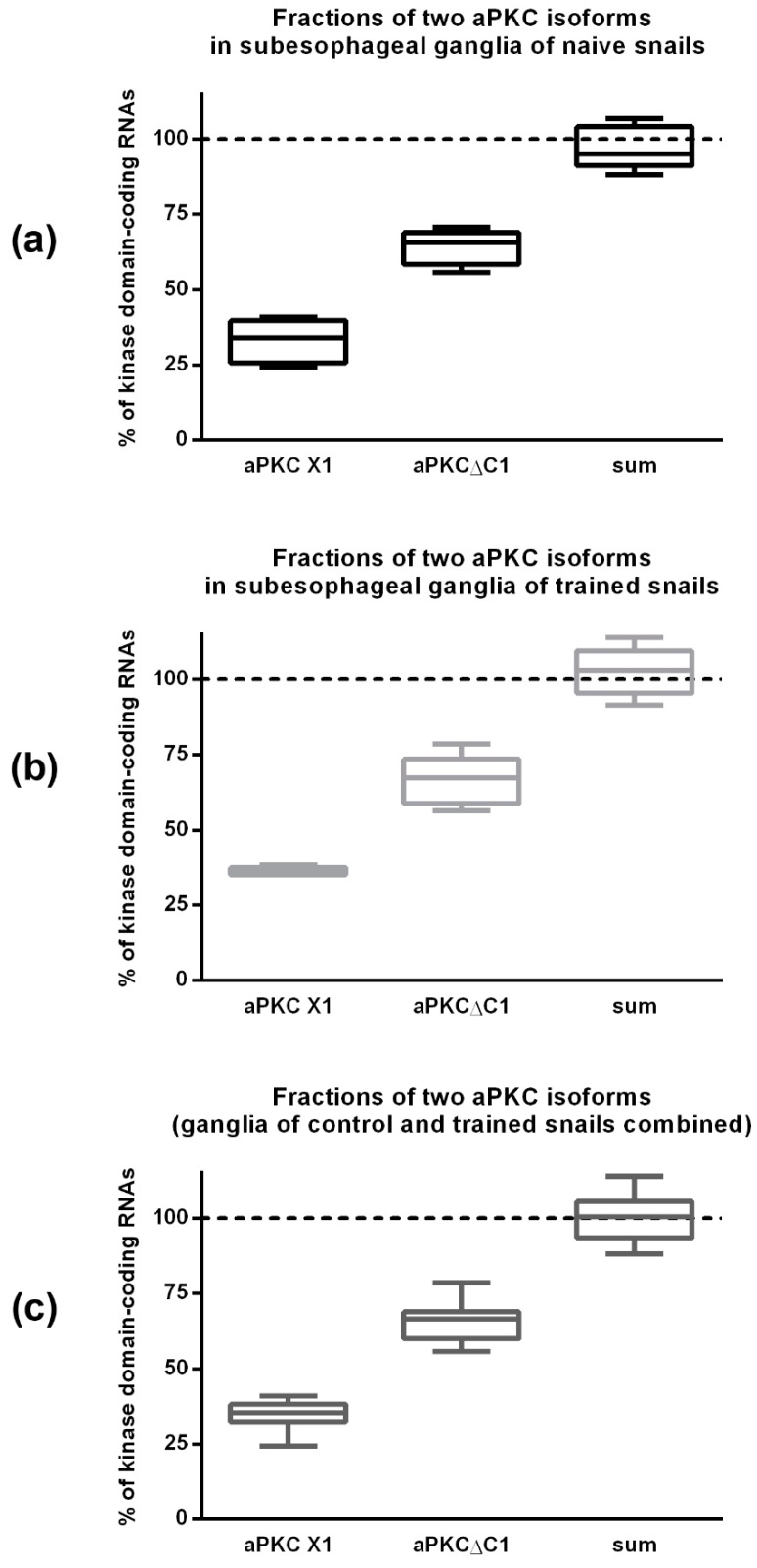
The number of cDNA fragments corresponding to different aPKC mRNA isoforms, normalized to the number of kinase domain-encoding cDNA fragments. The data was obtained by ddPCR using cDNA made from RNA extracted from subesophageal ganglia after contextual fear conditioning experiment. (**a**) Control (naive) snails, *n* = 5; (**b**) trained snails, *n* = 5; and (**c**) combined data for all 10 samples. Boxes represent median and quartiles; whiskers represent minimum and maximum values. Horizontal dotted lines mark 100% level. Sum–the total amount of aPKCΔC1 and aPKC X1 molecules taken together.

**Figure 6 ijms-20-02117-f006:**
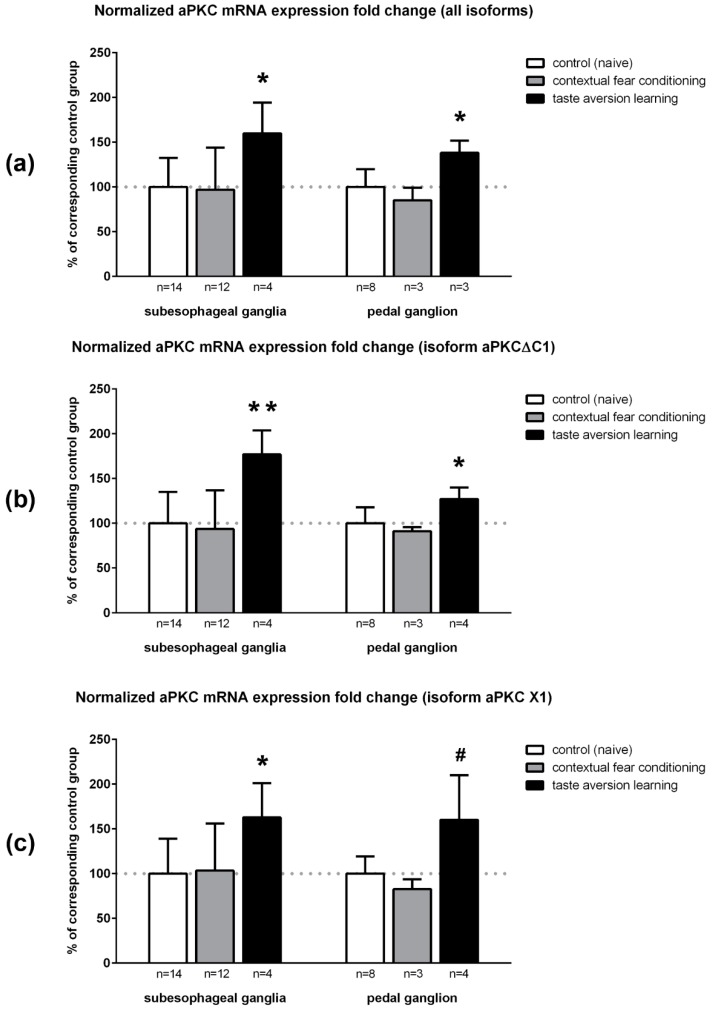
Normalized mRNA expression fold changes of total aPKC and two its isoforms in subesophageal and pedal ganglia of naive (control) snails and in two test groups of trained snails: (**a**) total aPKC; (**b**) aPKCΔC1 isoform; and (**c**) aPKC X1 isoform. Data are present as normalized AM ± SD. Horizontal dotted lines mark 100% level. * *p* < 0.05, ** *p* < 0.005, # *p* = 0.073 (Mann–Whitney test) compared to control.

**Table 1 ijms-20-02117-t001:** Gene-specific primers designed for putative snail aPKC mRNA transcript.

Primer Function	Primer Name	Primer Sequence (5′→3′)
Reverse primer for the 1st round of nested PCR	R1	GTTTGCACCCAGTCGATGTCC
Reverse primer for the 2nd round of nested PCR	R2	GACCAGCTCTTTCTTGATGACTTTC
Reverse primer for the 3rd round of nested PCR	R3	TGCAGCACCTTGGCGTAGC

**Table 2 ijms-20-02117-t002:** Gene-specific primers designed to assess expression levels of aPKC mRNA isoforms.

Primer Target	Primer Name	Primer Sequence (5′→3′)
aPKC X1 mRNA isoform	aPKC X1 F	CTGCATGTGCATTTTGCC
	aPKC X1 R	TTTCACATCCTCCAGTGTTCC
aPKCΔC1 mRNA isoform (primers for qPCR)	aPKCΔC1 F	TATAGGAGAGGGGCTCG
	aPKCΔC1 R	ATGTTCTTTCACATCCCTTG
aPKCΔC1 mRNA isoform (primers for ddPCR)	dd aPKCΔC1 F	GCCAAGAGATTTTCAAGGGAT
	dd aPKCΔC1 R	CTCCATTCACAGGTTGCG
Putative kinase domain-coding part of aPKC mRNA	aPKC KD F	TGAGTTTGTGAATGGAGGCG
	aPKC KD R	AGTCTGTTAGTTTGATGTGTCCC
TATA-box binding protein	TBP F	GGTTGGTAGCTGTGATGTC
	TBP R	CCATGCGGTAGATAAGTCC
Glyceraldehyde 3-phosphate dehydrogenase	GAPDH F	CCCAGAACATCATTCCCTCCTC
	GAPDH R	CGGAAAGCCATGCCGGT

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
