# Peer review of "Experiments with Snails Add to Our Knowledge about the Role of aPKC Subfamily Kinases in Learning"

_ijms, 2019, doi:10.3390/ijms20092117_

Round 1

Reviewer 1 Report

The introduction is very large and looks more like a review, and not an original study. The introduction should only provide the context and the objective of the research. It should be modified considerably to make the readers understand the concepts and not get them bored.

I think also the discussion section could be shorten. It should present only the future directions and comparation with similar studies. This section should not repeat that already presented in the results section

Author Response

The authors sincerely thank all 3 reviewers for their constructive comments regarding our manuscript. 

We have appreciated your suggestions and changed the text accordingly. Please examine the new version of the text, the file name is “ijms-481979-shortened-larger-pictures”. We used the "Track Changes" function in Microsoft Word to make all the changes visible.

We were advised to significantly shorten the introduction and discussion chapters and we did so by removing many redundant details.

Reviewer 2 Report

     This is a sound and well made work that makes interesting contributions to the field of PKCs protein quinases. Variant homologues of the atypical (aPKC) subfamily from the land snail Helix lucorum have been characterized at the transcriptomic level, cDNAs derived and sequenced, and the main elements of component domains, subdomains and isoforms characterized.  Also, the involvements in neurological roles, and more particularly in memory/learning and behaviour have been analysed and confirmed experimentally in such context.  The work appears as very carefully planned and conducted, with the proper experimental and theoretical approaches, as well as the corresponding statistical studies and proper controls, establishing well selected comparisons with equivalent systems in both invertebrates and vertebrates, leading overall to trustable conclusions.

     Some details that would require improvement are indicated below:

-The information provided on the transcriptome assembly of Helix lucorrum is vey scarce and, althought it is mentioned that has been conducted by a collaborative group, it would not allow repetition or taken as example by any reader, if wished. It should be increased.  Also, it would be correct to provide additional information on such a collaborative group (i.e. location/affiliation in the same or institutions other than those of the authors, to facilitate contacts if required).

-The lettering and numbering used in Fig.1 is at the limit of understanding by average readers (i.e. because of the small size or selected colour) and should be changed.  This is in contrast with the other Figures, very clearly labelled.

-The Supplemental figures are included in both the main manuscript and in the Supplemental attachment.  Only one is required.  By the way, the boxed start sites should be explained in the figure legend (Fig. A1).

Author Response

The authors sincerely thank all 3 reviewers for their constructive comments regarding our manuscript. 

The pictures with alignments were too small and hard to read, so we revised them. To make them larger and more legible, we divided Figure 1 into three separate figures and split Figure A1 into two fragments that fit in the page. We also added the explanation about the boxed start sites in the legends of Figure 1 and Figure A1.

We were also asked to provide more information about our collaborators' work with snail transcriptome assembly so we added the reference to the paper about this that will soon be published. The paper of P. Khaitovich with coauthors from this manuscript Balaban P., Aseyev N., Kolosov P. titled "Identification of immediate early genes in the nervous system of snail Helix lucorum" was recently accepted and sent to production in the journal eNeuro. We also provide supplementary text files containing sequences of the two snail contigs corresponding to aPKC mRNA that we used to design our experiments.

Reviewer 3 Report

In this manuscript, the authors established the sequence of mRNA encoding PKMζ homolog in land snail Helix lucorum and discovered this mRNA has at least two isoforms, and one of them lacks sequence encoding C1 domain. And they confirmed the relationship of identified isoforms with one type of learning by checking protein expression level in Helix model. This is a new discovery in molecular level based on their many years’ studies on animal model. The paper is well written, and easy to read. So, this manuscript could be considered for publication in International Journal of Molecular Sciences after minor modification, like,

1)     Figure 1, especially 1d should be modified due to low resolution.

Author Response

The authors sincerely thank all 3 reviewers for their constructive comments regarding our manuscript. 

The pictures with alignments were too small and hard to read, so we revised them. To make them larger and more legible, we divided Figure 1 into three separate figures and split Figure A1 into two fragments that fit in the page.